# Efficient Distillation of Classifier-Free Guidance using Adapters

**Cristian Perez Jensen**\*                                          *cjense@ethz.ch*
*ETH Zürich*

**Seyedmorteza Sadat**\*                                          *ssadat@ethz.ch*
*ETH Zürich*

**Reviewed on OpenReview:** *https://openreview.net/forum?id=uMz8FiiW01*

## Abstract

While classifier-free guidance (CFG) is essential for conditional diffusion models, it doubles the number of neural function evaluations (NFEs) per inference step. To mitigate this inefficiency, we introduce adapter guidance distillation (AGD), a novel approach that simulates CFG in a single forward pass. AGD leverages lightweight adapters to approximate CFG, effectively doubling the sampling speed while maintaining or even improving sample quality. Unlike prior guidance distillation methods that tune the entire model, AGD keeps the base model frozen and only trains minimal additional parameters (∼2%) to significantly reduce the resource requirement of the distillation phase. Additionally, this approach preserves the original model weights and enables the adapters to be seamlessly combined with other checkpoints derived from the same base model. We also address a key mismatch between training and inference in existing guidance distillation methods by training on CFG-guided trajectories instead of standard diffusion trajectories. Through extensive experiments, we show that AGD achieves comparable or superior FID to CFG across multiple architectures with only half the NFEs. Notably, our method enables the distillation of large models (∼2.6B parameters) on a single consumer GPU with 24 GB of VRAM, making it more accessible than previous approaches that require multiple high-end GPUs. We will publicly release the implementation of our method.

## 1 Introduction

Score-based diffusion models (Sohl-Dickstein et al., 2015; Ho et al., 2020; Song et al., 2020b) are a family of generative models that learn the data distribution by reversing a forward process that progressively corrupts the data until it becomes indistinguishable from pure Gaussian noise. Theoretically, running the reverse diffusion process should enable accurate sampling from the data distribution, assuming access to the ground truth score function. However, in practice, unguided sampling from diffusion models often produces low-quality images that fail to align well with the given input condition due to optimization errors (Karras et al., 2024a). Accordingly, CFG (Ho & Salimans, 2022) has become a crucial technique in modern conditional diffusion models to enhance both generation quality and alignment to conditioning signals—though this comes at the expense of reduced sample diversity (Ho & Salimans, 2022; Sadat et al., 2024).

CFG is an inference method that enhances generation quality by leveraging the difference between conditional and unconditional model predictions at each inference step. This difference serves as an update direction to improve both quality and alignment with the target condition. However, CFG requires two forward passes per inference step, resulting in twice the sampling cost compared to unguided sampling. This increased cost

---

\*These authors contributed equally to this work.

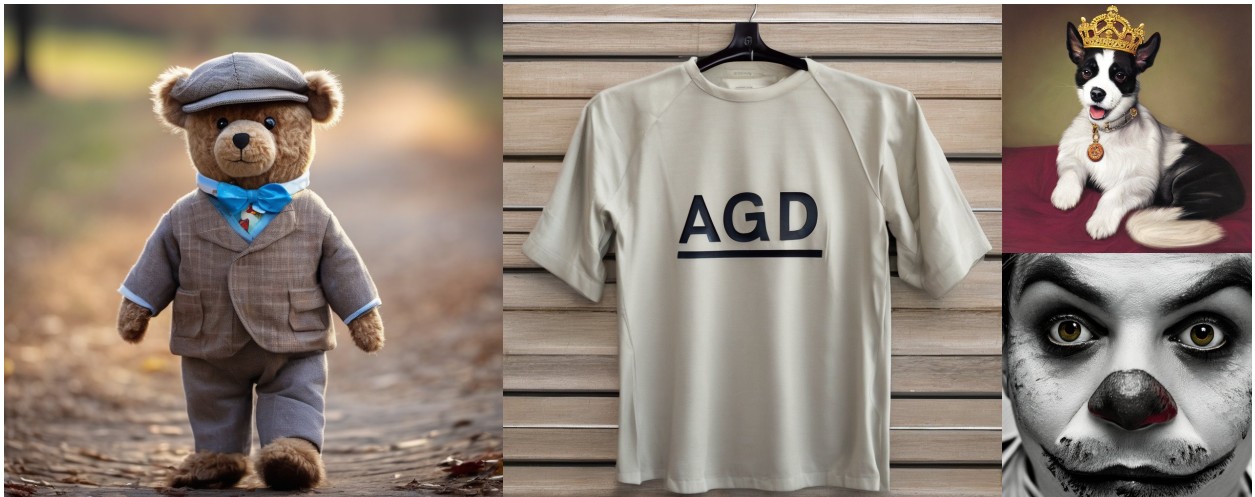

*Stable Diffusion XL*        *Stable Diffusion 2.1*

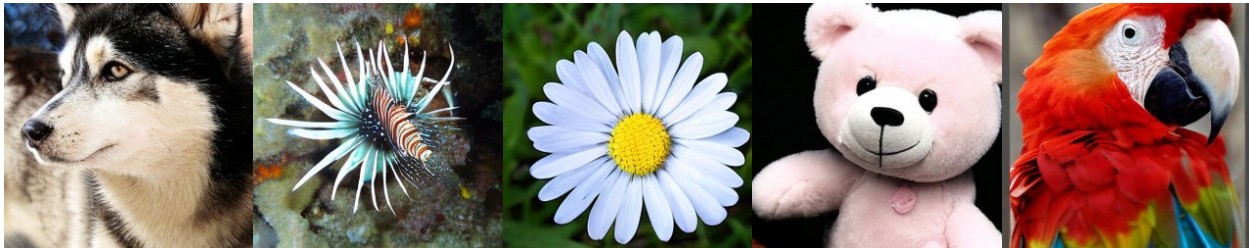

*Diffusion Transformer*

Figure 1: Generated samples using adapter guidance distillation (AGD) applied to various models. By efficiently baking classifier-free guidance (CFG) into the base diffusion model, AGD generates high-quality samples with only a single forward pass per inference step, which results in doubling the sampling speed compared to standard CFG.

introduces a significant computational overhead, especially when sampling from large-scale diffusion models or employing these pretrained models for tasks such as score distillation (Poole et al., 2023).

In this paper, we aim to double the sampling speed of CFG by training a small set of adapters to integrate the CFG behavior directly into the model. Our method, called AGD, learns to replicate the CFG-guided output at each inference step using a single forward pass while preserving the original diffusion model weights. These lightweight adapters add only 1–5% more parameters to the base model and introduce negligible latency overhead during inference. Since the base model remains frozen during training, and only the adapter parameters are updated, AGD is computationally efficient and can be trained on a single consumer GPU with 24 GB of VRAM, even for large models like Stable Diffusion XL (SDXL). Furthermore, AGD allows the trained adapters to be seamlessly integrated with other checkpoints originating from the same base model, such as IP-adapters (Ye et al., 2023). We demonstrate that our approach maintains or improves generation quality compared to standard CFG and outperforms existing methods such as guidance distillation (GD) (Meng et al., 2023), all while significantly reducing resource requirements during training.

Moreover, we identify and address a mismatch between training and inference trajectories in prior guidance distillation methods. We argue that effective guidance distillation requires training on *CFG-guided trajectories* computed by running the sampling process with CFG, as these differ significantly from standard diffusion trajectories obtained by adding noise to the training data. Furthermore, training on guided trajectories eliminates the need to load a teacher model during distillation, thus reducing memory requirements when training AGD. Another advantage is that the distillation can be performed entirely on the synthetic data

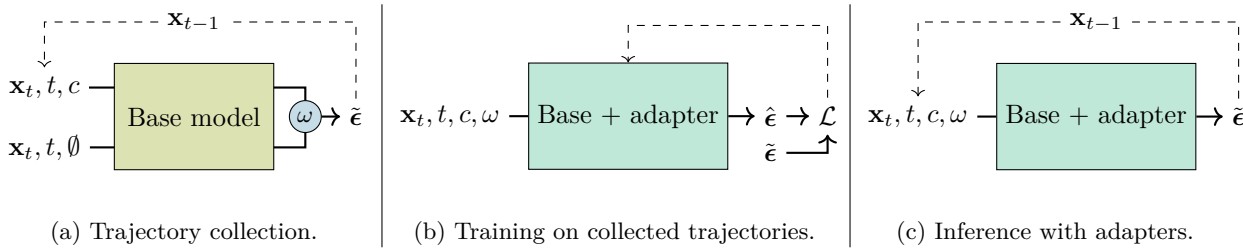

(a) Trajectory collection.     (b) Training on collected trajectories.     (c) Inference with adapters.

Figure 2: High-level overview of AGD components. (a) Instead of training on diffusion trajectories, we first run the sampling process with classifier-free guidance (CFG) and use the resulting guided trajectories (*i.e.*, intermediate predictions at each time step $t$) as our training dataset. (b) We then introduce small adapters to the base model and train them to replicate the CFG-guided predictions from (a) while keeping the base model frozen. (c) During inference, the base model combined with the trained adapter produces guided predictions in a single forward pass, effectively doubling the sampling speed compared to CFG.

generated by the teacher model without needing any real dataset in advance. Our experiments demonstrate that training on CFG-guided trajectories enhances performance compared with training on diffusion trajectories.

Figure 2 gives an overview of different components in AGD. In summary, our primary contributions are:

1. We introduce AGD, an efficient method for simulating CFG in a single forward pass by training lightweight adapters alongside a frozen base diffusion model, eliminating the need to fine-tune the entire model.

2. We propose training AGD on CFG-guided trajectories instead of diffusion trajectories, reducing the mismatch between training and inference and improving performance.

3. We demonstrate the resource efficiency of AGD by successfully distilling SDXL (2.6B parameters) on a single RTX 4090 GPU with 24 GB of VRAM.

4. Through extensive experiments, we show that AGD matches or surpasses CFG in performance across various models such as Diffusion Transformer and Stable Diffusion while doubling the sampling speed compared to CFG.

## 2 Related work

Diffusion models (Sohl-Dickstein et al., 2015; Song & Ermon, 2019; Song et al., 2020a;b; Ho et al., 2020) have emerged as a leading approach for generative modeling across various domains, including images (Ramesh et al., 2022; Rombach et al., 2022; Dai et al., 2023; Saharia et al., 2022), text (Hoogeboom et al., 2021; Li et al., 2022; Austin et al., 2021), audio (Evans et al., 2024), and molecular generation (Hoogeboom et al., 2022). Since the introduction of DDPM (Ho et al., 2020), significant progress has been made in multiple aspects, such as refining network architectures (Hoogeboom et al., 2023; Karras et al., 2024b; Peebles & Xie, 2023; Dhariwal & Nichol, 2021), developing more efficient sampling methods (Song et al., 2020a; Karras et al., 2022; Liu et al., 2022; Lu et al., 2022; Salimans & Ho, 2022b), and introducing novel training strategies (Nichol & Dhariwal, 2021; Song et al., 2020b; Salimans & Ho, 2022a; Rombach et al., 2022; Karras et al., 2022; 2024b). Nevertheless, various guidance techniques (Dhariwal & Nichol, 2021; Ho & Salimans, 2022; Karras et al., 2024a; Sadat et al., 2025b) have remained essential for enhancing generation quality and the alignment between conditioning inputs and generated outputs (Nichol et al., 2022), though they lead to increased sampling time (Ho & Salimans, 2022), reduced diversity (Sadat et al., 2024; Kynkäänniemi et al., 2024), and excessive oversaturation (Sadat et al., 2025a).

Several works have recently explored modifying the weight schedule of CFG by applying guidance only at certain sampling steps (Castillo et al., 2023; Wang et al., 2024; Kynkäänniemi et al., 2024), primarily to balance diversity and quality in generation. However, these methods still require two NFEs for most steps

and therefore cannot fully double the inference speed. Additionally, our approach is orthogonal to these methods, as the distilled model can be used for the steps where CFG is applied in the above works.

Alternatively, Meng et al. (2023) introduced guidance distillation (GD), which fine-tunes a diffusion model to generate guided predictions in a single forward pass. However, fully fine-tuning the base model is often inefficient and unstable for large models, overwrites the original model weights, and demands high-end GPUs with substantial memory for training. To improve efficiency of GD, Hsiao et al. (2024) introduced plug-and-play diffusion distillation (PPDD), which injects residuals via an auxiliary model. However, their most accurate variant (requiring approximately 42% additional parameters) still underperforms CFG by roughly 14% in FID. To address these limitations, we propose a more efficient distillation method using adapters, which achieves substantially better performance while remaining significantly more parameter-efficient. Moreover, both Meng et al. (2023) and Hsiao et al. (2024) trained their models on standard diffusion trajectories, which we show to be less effective than training the distillation process on CFG-guided trajectories.

Finally, adapters (Houlsby et al., 2019) have emerged as a parameter-efficient solution for fine-tuning large-scale diffusion models, mainly for integrating image conditions into pretrained text-to-image models (Mou et al., 2024; Ye et al., 2023). In contrast, we leverage adapters to inject guided predictions directly into the model's forward pass. Notably, we demonstrate that adapters not only require significantly fewer training resources but also slightly outperform full fine-tuning.

## 3 Background

In this section, we provide a concise overview of diffusion models. Consider a data point $\mathbf{x} \sim p_{\text{data}}$ and noise $\boldsymbol{\epsilon} \sim \mathcal{N}(\mathbf{0}, \mathbf{I})$, and let the forward diffusion process be defined as $\mathbf{x}_t = \mathbf{x} + \sigma(t)\boldsymbol{\epsilon}$, where noise is gradually introduced over time $t \in [0, T]$. The function $\sigma(t)$ serves as the noise schedule, determining the extent of perturbation at each step, with $\sigma(0) = 0$ and $\sigma(T) = \sigma_{\text{max}}$. As shown by Karras et al. (2022), this process can be described by the following ordinary differential equation (ODE):

$$\mathrm{d}\mathbf{x}_t = -\dot{\sigma}(t)\sigma(t)\,\nabla_{\mathbf{x}_t} \log p_t(\mathbf{x}_t)\mathrm{d}t, \tag{1}$$

where $p_t$ is the marginal distribution of the noisy data at time step $t$, transitioning from the original data distribution $p_0 = p_{\text{data}}$ to a Gaussian prior $p_T = \mathcal{N}(\mathbf{0}, \sigma_{\text{max}}^2\mathbf{I})$.

Assuming access to the time-dependent score function $\nabla_{\mathbf{x}_t} \log p_t(\mathbf{x}_t)$, one can solve this ODE in reverse, *i.e.*, from $t = T$ to $t = 0$, to generate new samples from $p_{\text{data}}$. The unknown score function $\nabla_{\mathbf{x}_t} \log p_t(\mathbf{x}_t)$ is typically learned using a neural denoiser $D_{\boldsymbol{\theta}}(\mathbf{x}_t, t)$, which is trained to recover clean samples $\mathbf{x}$ from their noisy counterparts $\mathbf{x}_t$. Additionally, conditional generation can be achieved by extending the denoiser to $D_{\boldsymbol{\theta}}(\mathbf{x}_t, t, c)$, where $c$ represents auxiliary conditioning information, such as class labels or text.

**Training** Following (Ho et al., 2020), the denoiser $D_{\boldsymbol{\theta}}(\mathbf{x}_t, t)$ is commonly parameterized as

$$D_{\boldsymbol{\theta}}(\mathbf{x}_t, t) = \mathbf{x}_t - \sigma(t)\boldsymbol{\epsilon}_{\boldsymbol{\theta}}(\mathbf{x}_t, t), \tag{2}$$

and is trained by predicting the added noise $\boldsymbol{\epsilon}$ in $\mathbf{x}_t$, that is by solving

$$\boldsymbol{\theta}^{\star} \in \operatorname*{argmin}_{\boldsymbol{\theta}} \mathbb{E}_{\mathbf{x}, \boldsymbol{\epsilon}, t}\left[\|\boldsymbol{\epsilon}_{\boldsymbol{\theta}}(\mathbf{x}_t, t) - \boldsymbol{\epsilon}\|^2\right]. \tag{3}$$

After training, the score function can be approximated via

$$\nabla_{\mathbf{x}_t} \log p_t(\mathbf{x}_t) \approx \frac{D_{\boldsymbol{\theta}}(\mathbf{x}_t, t) - \mathbf{x}_t}{\sigma(t)^2} = -\frac{\boldsymbol{\epsilon}_{\boldsymbol{\theta}}(\mathbf{x}_t, t)}{\sigma(t)}. \tag{4}$$

**Classifier-free guidance** CFG is an inference technique aimed at improving the quality of generated samples by blending the outputs of a conditional and an unconditional model (Ho & Salimans, 2022). Specifically, CFG adjusts the denoiser's output at each sampling step according to

$$\tilde{\boldsymbol{\epsilon}}_{\boldsymbol{\theta}}(\mathbf{x}_t, t, c, \omega) = \omega\boldsymbol{\epsilon}_{\boldsymbol{\theta}}(\mathbf{x}_t, t, c) - (\omega - 1)\boldsymbol{\epsilon}_{\boldsymbol{\theta}}(\mathbf{x}_t, t, \emptyset), \tag{5}$$

---

**Algorithm 1** Trajectory collection for AGD.

---

**Require:** Set of conditions $\mathcal{C}$, Guidance scale range $[\omega_{\min}, \omega_{\max}]$, Pretrained diffusion model $\epsilon_{\boldsymbol{\theta}}$
  1: $\Omega = \emptyset$
  2: **for** $c \in \mathcal{C}$ **do**
  3:     $\omega \sim \text{Unif}([\omega_{\min}, \omega_{\max}])$
  4:     Run the reverse diffusion process from Equation (1) using $\epsilon_{\boldsymbol{\theta}}$ with $\omega$ and $c$ in $N$ steps
  5:     Cache $(\mathbf{x}_t, t, c, \omega)$ and the CFG-guided prediction $\tilde{\epsilon}_{\boldsymbol{\theta}}(\mathbf{x}_t, t, c, \omega)$ at each sampling step:
       $\Omega \leftarrow \Omega \cup \{(\mathbf{x}_t, t, c, \omega, \tilde{\epsilon}_{\boldsymbol{\theta}}(\mathbf{x}_t, t, c, \omega))\}_{t=1}^{N}$
  6: **end for**

---

**Algorithm 2** Adapter training for AGD.

---

**Require:** Trajectory dataset $\Omega$ from Algorithm 1, Model with adapters $\epsilon_{[\boldsymbol{\theta}, \boldsymbol{\psi}]}$, Loss function $\ell$
  1: **while** not converged **do**
  2:     $(\mathbf{x}_t, t, c, \omega, \tilde{\epsilon}) \sim \text{Unif}(\Omega)$
  3:     $\mathcal{L} = \ell(\tilde{\epsilon}, \epsilon_{[\boldsymbol{\theta}, \boldsymbol{\psi}]}(\mathbf{x}_t, t, c, \omega))$
  4:     Update $\boldsymbol{\psi}$ with gradient step on $\mathcal{L}$
  5: **end while**

---

where $\omega = 1$ corresponds to unguided sampling, and $c = \emptyset$ represents the unconditional prediction. The unconditional model $\epsilon_{\boldsymbol{\theta}}(\mathbf{x}_t, t, \emptyset)$ is typically trained by randomly replacing the conditioning input with $c = \emptyset$ during training. Alternatively, a dedicated denoiser can be trained separately to approximate the unconditional score (Karras et al., 2024b). CFG is known to significantly improve generation quality, though it comes at the cost of doubling the sampling time (Ho & Salimans, 2022).

## 4 Adapter guidance distillation

We now introduce our method, adapter guidance distillation (AGD), for doubling the sampling speed of CFG. As shown in Figure 2, AGD consists of two main components: (1) training on CFG-guided trajectories instead of standard diffusion trajectories, and (2) training lightweight adapters to distill CFG instead of fine-tuning the full model. Below, we discuss each component in detail, with Algorithms 1 and 2 also providing the training details of AGD.

### 4.1 Training on CFG-guided trajectories

Prior guidance distillation methods are trained on standard diffusion trajectories, where noise is added to the training data, and the CFG prediction is matched at each inference step (Meng et al., 2023). However, since CFG modifies the reverse process of diffusion models, guided trajectories differ significantly from standard diffusion trajectories, as shown in Figure 3. We argue that training directly on CFG-guided trajectories enhances guidance distillation by exposing the model to regions in space that the guided reverse process will follow. To bridge the gap between training and inference, we thus train AGD directly on CFG-guided trajectories. We generate guided trajectories as outlined in Algorithm 1, which are then used to train AGD. Since these trajectories can be cached, the teacher model does not need to be loaded during training, freeing up VRAM. Moreover, because this method only depends on generated samples from the teacher model, it does not require an external dataset for training. Additionally, the trajectory dataset only needs to be collected once, enabling efficient hyperparameter tuning for the adapters.

### 4.2 Efficient guidance distillation with adapters

For more efficient training, AGD only uses small learnable modules, or *adapters* (Houlsby et al., 2019), to replicate the effect of CFG. Unlike tuning the whole diffusion network as in GD (Meng et al., 2023), we freeze the original model weights, ensuring that the base model is still available after training. This also allows

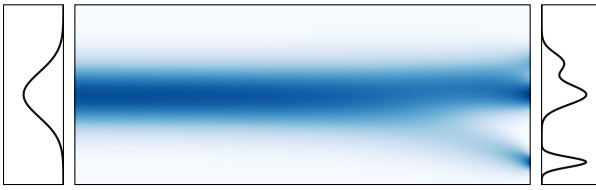

(a) Visualizing the marginal densities of the diffusion process with a mixture of Gaussians as the data distribution.

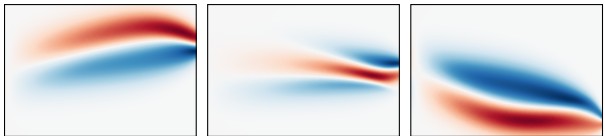

(b) Illustration of the distribution mismatch between conditional diffusion (blue) and classifier-free guidance (red) trajectories during the reverse diffusion process.

Figure 3: One-dimensional illustration of the mismatch between standard diffusion trajectories used for training in existing guidance distillation methods (such as GD (Meng et al., 2023)) and the actual CFG-guided trajectories followed during inference.

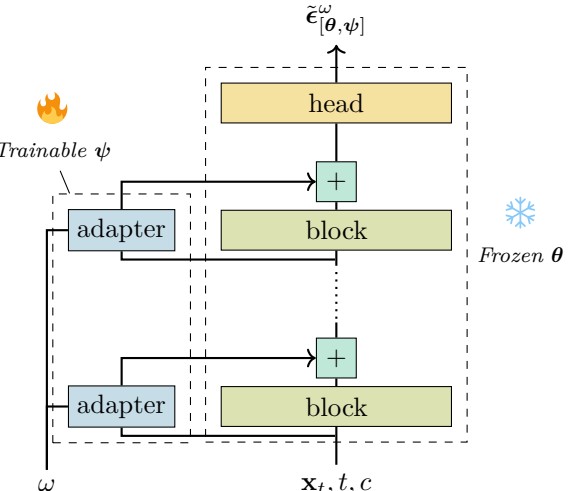

Figure 4: Visual illustration of the trainable adapters alongside the frozen base model. The adapters are typically integrated with attention layers (either self-attention or cross-attention), and their outputs are added to those of the frozen attention blocks.

us to use the learned adapters with other checkpoints that are obtained from the same base model, such as IP-adapters (Ye et al., 2023). The details of the adapters used in AGD are given below.

**Adapter formulation** Let $f_{\boldsymbol{\theta}}$ denote an intermediate layer in the network with $\mathbf{Z} \in \mathbb{R}^{L \times d}$ being its upstream input. Further, $f_{\boldsymbol{\theta}}$ receives the time step $t$ and the condition embedding $c$ as input. An adapter $g_{\boldsymbol{\psi}}$ with parameters $\boldsymbol{\psi}$ is a layer that combines $f_{\boldsymbol{\theta}}$ with encodings of the guidance scale $\omega$, and the input conditions $(t, c)$ via residual connection:

$$\tilde{f}_{[\boldsymbol{\theta}, \boldsymbol{\psi}]}(\mathbf{Z}, \omega, t, c) = f_{\boldsymbol{\theta}}(\mathbf{Z}, t, c) + g_{\boldsymbol{\psi}}(\mathbf{Z}, \omega, t, c). \tag{6}$$

This architecture is illustrated in Figure 4. During training, the model weights $\boldsymbol{\theta}$ are kept frozen and only the adapter parameters $\boldsymbol{\psi}$ are optimized to match the CFG step based on the trajectory dataset, as introduced in Section 4.1, *i.e.*,

$$\boldsymbol{\psi}^{\star} \in \underset{\boldsymbol{\psi}}{\operatorname{argmin}} \, \mathbb{E}\big[\ell(\boldsymbol{\epsilon}_{[\boldsymbol{\theta}, \boldsymbol{\psi}]}(\mathbf{x}_t, t, c, \omega), \tilde{\boldsymbol{\epsilon}}_{\boldsymbol{\theta}}(\mathbf{x}_t, t, c, \omega))\big], \tag{7}$$

where $\tilde{\boldsymbol{\epsilon}}_{\boldsymbol{\theta}}(\mathbf{x}_t, t, c, \omega)$ denotes a CFG step with guidance scale $\omega$, $\boldsymbol{\epsilon}_{[\boldsymbol{\theta}, \boldsymbol{\psi}]}(\mathbf{x}_t, t, c, \omega)$ is the output of the model with the adapters, and $\ell$ is the loss function.

**Adapter architecture** We mainly experiment with two adapter architectures: (1) cross-attention adapters, and (2) offset adapters. Let $\mathbf{C} = [\mathbf{c}_1, \ldots, \mathbf{c}_C]$ represent the matrix containing all conditioning embeddings (*e.g.*, guidance scale, prompt embeddings, *etc.*), linearly projected to the same dimensionality via a learned projection. Akin to IP-adapter (Ye et al., 2023), the cross-attention adapter formulates $g_{\boldsymbol{\psi}}$ as

$$g_{\boldsymbol{\psi}}(\mathbf{Z}, \omega, t, c) = \operatorname{Softmax}\left(\frac{\mathbf{Q}\mathbf{K}^{\top}}{\sqrt{d}}\right)\mathbf{V}, \tag{8}$$

where $\mathbf{Q} = \mathbf{Z}\mathbf{W}_q$, $\mathbf{K} = \mathbf{C}\mathbf{W}_k$, and $\mathbf{V} = \mathbf{C}\mathbf{W}_v$. The offset adapter formulates $g_{\boldsymbol{\psi}}$ as

$$g_{\boldsymbol{\psi}}(\mathbf{Z}, \omega, t, c) = \operatorname{MLP}\left(\sum_{i=1}^{C} \mathbf{c}_i\right) \tag{9}$$

We found that offset adapters perform better for simpler models like DiT, whereas cross-attention adapters are more effective for text-to-image models. Several ablations on other adapter design space are provided in Appendix B.

**Implementation details**  We embed the guidance scale $\omega$ via a Fourier feature encoder (Tancik et al., 2020) followed by na multi-layer perceptron (MLP). We also extract the text or class embeddings from the base model (*e.g.*, CLIP embeddings) and linearly project them into the same dimensionality as the guidance scale embedding. In DiT (Peebles & Xie, 2023), we place an adapter in each transformer block after the self-attention mechanism. For text-to-image models such as Stable Diffusion 2.1 (SD2.1) (Rombach et al., 2022) and SDXL (Podell et al., 2024), we place the adapters in conjunction with the cross-attention layers, since the text prompt is only used in these blocks.

**Efficiency**  Since the adapters introduce only 1–5% additional parameters relative to the base model, their computational overhead remains negligible during both training and inference. Furthermore, unlike CFG, which requires *two* forward passes per diffusion step, our approach performs only *one*, effectively reducing the NFEs by half. Consequently, our method achieves twice the speed of CFG when generating samples from pretrained diffusion models.

## 5 Experiments and results

**Setup**  We evaluate AGD on class-conditional generation using 256×256 DiT-XL/2 (Peebles & Xie, 2023), and text-to-image generation using 768×768 Stable Diffusion 2.1 (SD2.1) (Rombach et al., 2022) and 1024×1024 Stable Diffusion XL (SDXL) (Podell et al., 2024). All experiments are conducted on a single RTX 4090 GPU (24 GB of VRAM). Training is performed using the Adam optimizer (Kingma & Ba, 2014) without weight decay, where the learning rate follows a linear warm-up to $1 \times 10^{-4}$ over the first 10% of steps, after which it decays via a cosine annealing schedule (Loshchilov & Hutter, 2016). For training adapters on DiT, trajectories are sampled with guidance scales $\omega \sim \text{Unif}([1, 6])$, with four trajectories per class label of ImageNet (Deng et al., 2009). For text-to-image models, we randomly select 500 captions from the COCO-2017 training set (Lin et al., 2014), generating a single trajectory per caption with guidance scales $\omega \sim \text{Unif}([1, 12])$. Please refer to Appendix D for additional details regarding the experiments.

**Evaluation metrics**  We mainly use FID (Heusel et al., 2017) to measure the quality and diversity of generated images, since it closely aligns with human perception. Given FID's sensitivity to implementation details, we evaluate all models under identical conditions to ensure consistency. Additionally, we report precision as a measure of quality and recall as an indicator of diversity (Kynkäänniemi et al., 2019).

### 5.1 Qualitative results

We evaluate the qualitative performance of AGD and CFG in Figure 5, generating samples using the same random seeds for both methods. Our results indicate that AGD produces images structurally similar to CFG while being more visually appealing across multiple models and resolutions. Thus, AGD retains the quality benefits of CFG while achieving twice the sampling speed per image.

### 5.2 Quantitative results

The quantitative evaluation of AGD and CFG is shown in Table 1. We observe that AGD achieves metrics comparable to CFG, with both methods significantly outperforming the unguided sampling baseline. This confirms that AGD enhances generation quality similarly to CFG while requiring only half the NFEs. Notably, AGD even slightly outperforms CFG for the DiT model.

### 5.3 Comparing AGD with guidance distillation

We next compare our method to guidance distillation (GD) (Meng et al., 2023), which fine-tunes the entire diffusion model to replicate guided outputs. We train AGD and GD under the same training setup using DiT

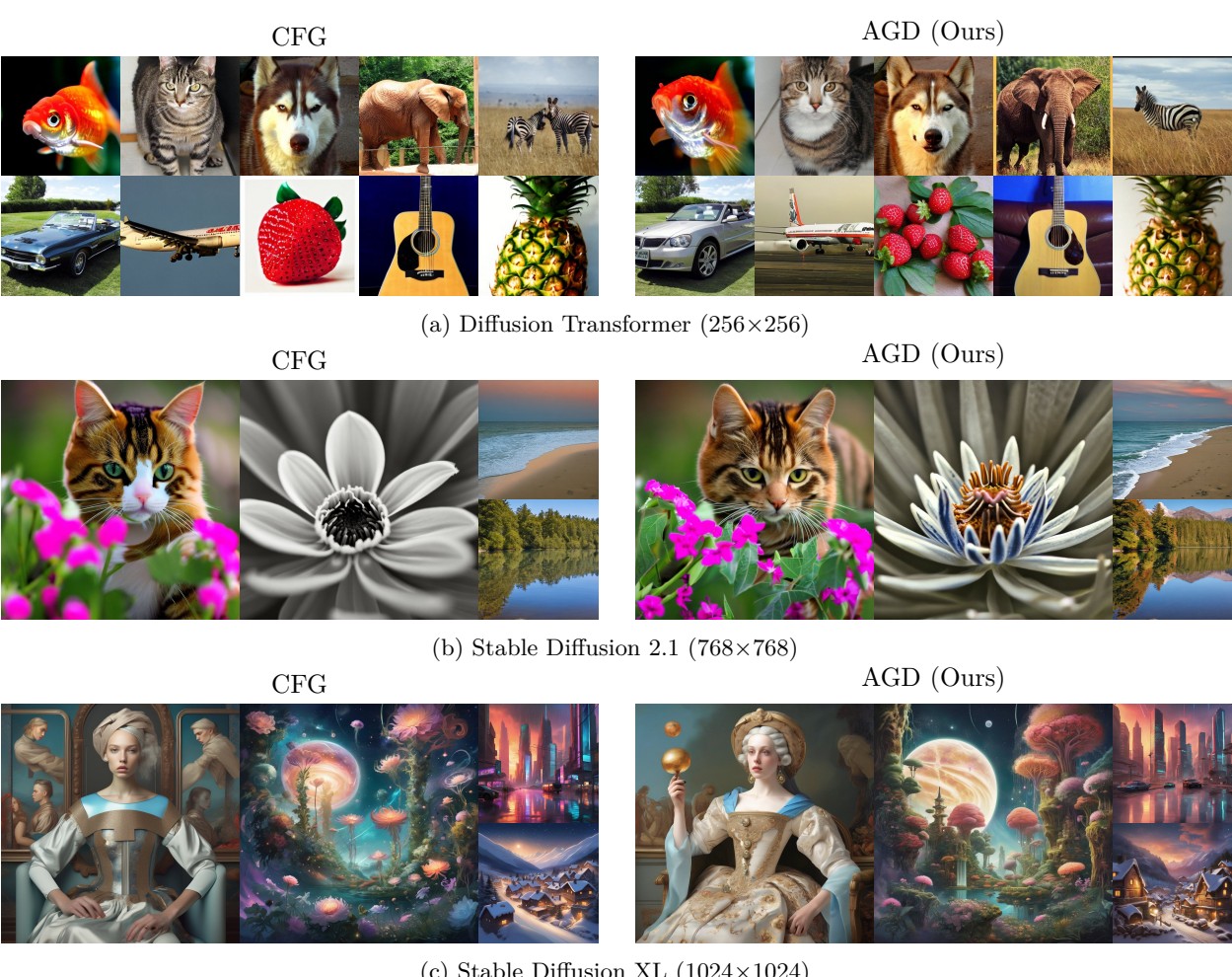

(a) Diffusion Transformer (256×256)

(b) Stable Diffusion 2.1 (768×768)

(c) Stable Diffusion XL (1024×1024)

Figure 5: Qualitative comparison between AGD and CFG. AGD produces samples with comparable quality to CFG while achieving twice the inference speed by requiring only a single forward pass through the model. Additionally, AGD samples maintain structural similarity to CFG but often have better visual coherence.

Table 1: Quantitative comparison between AGD and CFG. AGD outperforms CFG in class-conditional generation using DiT and performs similarly for text-to-image models (SD2.1 and SDXL). The throughput is computed with batch size 8.

| Model | Guidance | FID ↓ | Prec. ↑ | Recall ↑ | NFEs | Steps/s |
|---|---|---|---|---|---|---|
| DiT (Peebles & Xie, 2023) | Unguided | 12.57 | 0.67 | **0.74** | 250 | 15.84 |
| | CFG | 5.30 | **0.83** | 0.66 | 500 | 8.83 |
| | AGD (Ours) | **5.03** | 0.80 | 0.68 | 250 | 15.50 |
| SD2.1 (Rombach et al., 2022) | Unguided | 49.94 | 0.39 | **0.63** | 50 | 1.81 |
| | CFG | **20.94** | **0.67** | 0.55 | 100 | 0.91 |
| | AGD (Ours) | 21.09 | 0.66 | 0.55 | 50 | 1.79 |
| SDXL (Podell et al., 2024) | Unguided | 60.30 | 0.35 | **0.54** | 50 | 0.66 |
| | CFG | **22.82** | 0.66 | 0.52 | 100 | 0.33 |
| | AGD (Ours) | 22.98 | **0.67** | 0.52 | 50 | 0.64 |

as the base diffusion model for class-conditional ImageNet generation. Table 2 shows that AGD outperforms GD in FID while having significantly less trainable parameters. Thus, we conclude that GD can be made significantly more efficient by keeping the base model frozen and only training the adapters. Moreover,

Table 2: Comparing AGD and GD (Meng et al., 2023) using DiT under the same training setup. AGD slightly outperforms GD while only training the adapters instead of tuning the full model.

| Method | Params | FID ↓ | Prec. ↑ | Recall ↑ |
|---|---|---|---|---|
| GD | 676 M | 5.66 | **0.80** | 0.67 |
| AGD (Ours) | 16 M | **5.03** | **0.80** | **0.68** |

Table 3: Importance of training on guided trajectories. AGD performs best when trained on CFG-guided trajectories instead of the standard diffusion trajectories used in Meng et al. (2023).

| Method | FID ↓ | Prec. ↑ | Recall ↑ |
|---|---|---|---|
| CFG | 5.30 | **0.83** | 0.66 |
| AGD (Diffusion) | 5.54 | 0.80 | **0.68** |
| AGD (Trajectory) | **5.03** | 0.80 | **0.68** |

GD (Meng et al., 2023)    AGD (Ours)

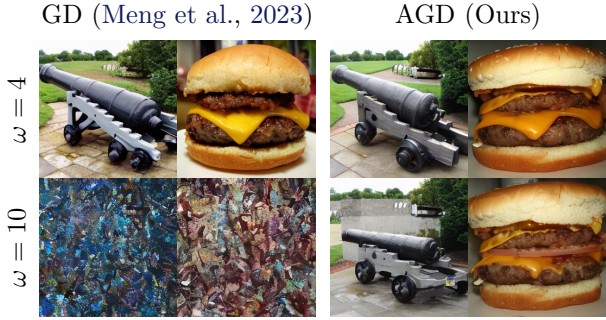

Figure 6: Comparison of AGD with guidance distillation (GD) (Meng et al., 2023) for unseen guidance scales. While GD fails completely for out-of-domain guidance scales, AGD continues to generate meaningful images. The models in this experiment were trained for $\omega \in [1,6]$.

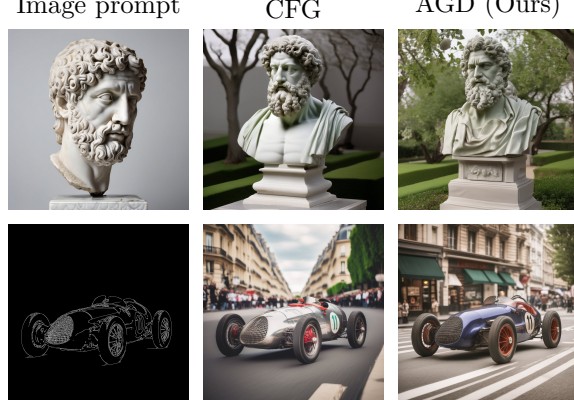

Figure 7: Using AGD with IP-adapter (Ye et al., 2023) and ControlNet (Zhang et al., 2023) for SDXL. AGD can be integrated with other checkpoints derived from the same base model, achieving the benefits of both modules.

Figure 6 shows that GD completely fails when used with guidance scales outside the domain seen during fine-tuning. In contrast, AGD remains robust to this issue, demonstrating better generalization across guidance scales.

## 5.4 Importance of training on guided trajectories

In order to validate our claim that training on CFG-guided trajectories is beneficial, we compared AGD trained on standard diffusion trajectories with AGD trained on guided trajectories. As shown in Table 3, training on guided trajectories leads to a substantial improvement over training on standard diffusion trajectories. Hence, we conclude that bridging the train-inference gap by aligning these trajectories enhances performance, as it focuses training on regions of the space that are important for CFG.

## 5.5 Combining AGD with other fine-tuned checkpoints

Figure 7 shows samples generated by combining AGD with IP-Adapter (Ye et al., 2023) and ControlNet (Zhang et al., 2023). As shown, AGD maintains high sample quality when integrated with other checkpoints derived from the same base model, enabling controllable generation at twice the sampling speed.

## 5.6 Training efficiency

Table 4 compares the training speed and VRAM usage of AGD and GD (Meng et al., 2023). We note that for larger networks like SDXL, AGD can successfully distill the model using a consumer GPU with 24 GB

Table 4: Comparing the memory requirements and training speed of AGD and GD. AGD enables the distillation of large models on an RTX 4090 with 24 GB of VRAM while also being significantly faster at each training iteration.

| Model | Method | VRAM (GB) | It/s |
|---|---|---|---|
| DiT (Peebles & Xie, 2023) | GD | 17.67 | 0.52 |
| | AGD (Ours) | 16.79 | 2.36 |
| SD2.1 (Rombach et al., 2022) | GD | *Out-of-memory* | |
| | AGD (Ours) | 23.83 | 2.05 |
| SDXL (Podell et al., 2024) | GD | *Out-of-memory* | |
| | AGD (Ours) | 22.77 | 3.94 |

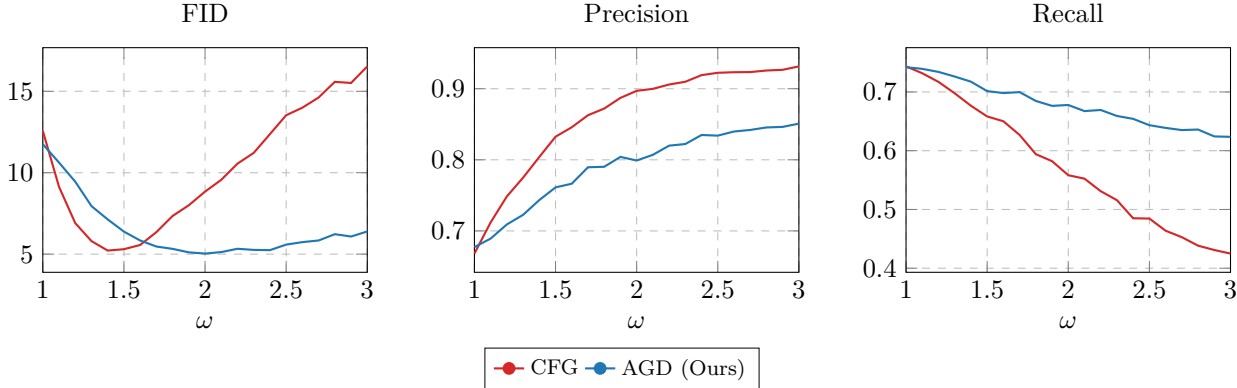

Figure 8: The performance of DiT with AGD as we increase the guidance scale. Compared to CFG, AGD offers a better trade-off between precision and recall resulting in better FID for most guidance scales.

VRAM, whereas GD encounters out-of-memory issues. Even when VRAM is not a constraint, each training step of AGD remains significantly more efficient than GD ($\sim 4.5\times$ faster for DiT).

### 5.7 AGD and guidance scale

Figure 8 shows how the performance of AGD varies as we increase the guidance scale $\omega$. We observe that the FID curve for CFG is more peaked, whereas the curve for AGD is relatively flatter, making it less sensitive to the exact guidance value at inference for good performance. Additionally, we note that AGD have a more favorable trade-off between precision and recall compared to CFG, resulting in better FID scores for most guidance scales.

### 5.8 Changing the scheduler at inference time

Next, we demonstrate that AGD is robust to the choice of scheduler for generating guided trajectories. In Figure 9, we show samples from the DDPM algorithm Ho et al. (2020) using adapters trained on DDIM trajectories (Song et al., 2020a). Even when a different scheduler is used at inference, AGD consistently produces high-quality images. Moreover, Table 5 reports comparable FID scores for both DDIM and DDPM. Note that the slight FID advantage of DDIM reflects its efficiency at lower sampling steps (e.g., 50), rather than any degradation in AGD.

Table 5: Compatibility of AGD with different diffusion samplers at inference.

| Sampler | FID ↓ | Prec. ↑ | Recall ↑ |
|---|---|---|---|
| DDIM | **21.09** | 0.66 | **0.55** |
| DDPM | 22.15 | **0.67** | 0.51 |

DDIM 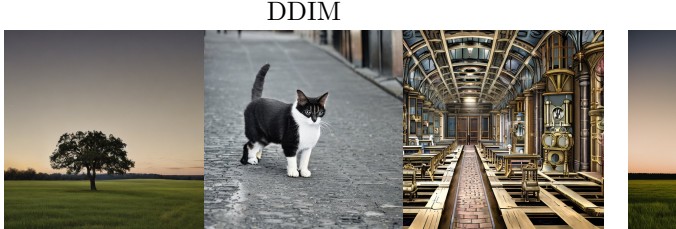 DDPM 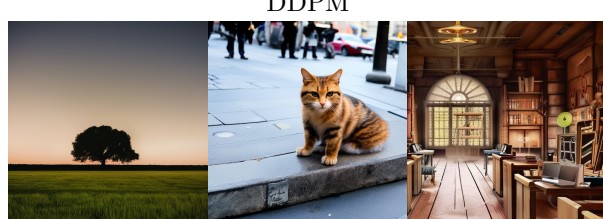

Figure 9: Samples of SD2.1 generated with AGD using the DDPM sampler, with adapters trained on DDIM trajectories. AGD successfully generates high-quality samples when used with different schedulers at inference.

## 5.9 Training and inference wall-clock times

To further demonstrate the computational efficiency of our method, we compared the wall-clock training and inference times of AGD and GD on DiT using an RTX 4090 GPU. For training, GD requires 160.3 minutes to complete, whereas AGD achieves comparable performance in just 33.9 minutes (*i.e.* a $4.7\times$ speedup). During inference, both AGD and unguided DiT require 6.0 seconds per sample, indicating that the adapter introduces negligible computational overhead. This confirms that the core advantage of AGD—reducing forward passes by half compared to CFG—translates directly into inference speedup without incurring any additional per-step costs. The combination of faster training, reduced memory usage, and accelerated inference makes AGD a compelling alternative to both standard CFG and existing guidance distillation approaches.

## 6 Conclusion

This paper introduced adapter guidance distillation (AGD), an efficient approach to achieving the benefits of classifier-free guidance at half the sampling cost. By training lightweight adapters to estimate guided outputs and training on CFG-guided trajectories, we address both the computational overhead and the train-inference mismatch of prior guidance distillation methods. Through extensive experiments, we showed that AGD matches or surpasses CFG's performance, remains robust to previously unseen guidance scales, and can be trained on a single consumer GPU even for large models such as SDXL. Thus, we believe that AGD offers an efficient and flexible alternative to prior guidance distillation methods while eliminating the sampling overhead of classifier-free guidance. Future research directions could explore integrating AGD with enhanced guidance algorithms (Kynkäänniemi et al., 2024; Karras et al., 2024a; Sadat et al., 2025a) and leveraging adapters for other distillation techniques, *e.g.*, adversarial distillation (Sauer et al., 2024; 2025), to further reduce the sampling time of diffusion models.

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

## A   Broader impact

Our method accelerates guided sampling in diffusion models, broadening accessibility to large-scale text-to-image or class-conditional generative systems. This can reduce energy consumption and computational barriers to use AI-generated content for various creative applications. However, while advancements in AI-generated content have the potential to improve efficiency and stimulate creativity, it is essential to consider the associated ethical implications. For a more in-depth exploration of ethics and creativity in computer vision, we refer readers to Rostamzadeh et al. (2021)

## B   Ablation studies

This section presents our ablation studies. Unless otherwise specified, all experiments are conducted using the DiT model for class-conditional generation. We use FID as the primary metric to determine the adapter configuration used in the main experiments.

**Adapter architecture**   We first examine various design choices for the adapter architecture $g_{\boldsymbol{\psi}}(\mathbf{Z}, \omega, t, c)$. Let $\mathbf{C} = [\mathbf{c}_1, \dots, \mathbf{c}_C]$ represent the matrix containing all conditioning embeddings. The cross-attention and offset adapter architectures are formalized in Equations (8) and (9) respectively. We further experimented with a gating architecture defined as

$$g_{\boldsymbol{\psi}}(\mathbf{Z}, \omega, t, c) = \big(\sigma(\tilde{\mathbf{Z}}\mathbf{v}) \odot \mathrm{MLP}(\tilde{\mathbf{Z}})\big)\mathbf{W}, \tag{10}$$

where $\tilde{\mathbf{z}}_j = \Big[\mathbf{z}_j, \sum_{i=1}^{C} \mathbf{c}_i\Big]$, $\sigma$ is the sigmoid function, and $\odot : \mathbb{R}^T \times \mathbb{R}^{T \times d} \to \mathbb{R}^{T \times d}$ scales each $d$-dimensional vector independently. Lastly, we also considered a positional encoding adapter architecture given by

$$g_{\boldsymbol{\psi}}(\mathbf{Z}, \omega, t, c) = \mathrm{MLP}(\tilde{\mathbf{c}}), \tag{11}$$

where $\tilde{\mathbf{c}}_j = \Big[\mathbf{e}_j, \sum_{i=1}^{C} \mathbf{c}_i\Big]$ and $\mathbf{e}_j$ encodes the $j$-th attention time step. Specifically, $\mathbf{e}_j$ is computed by a Fourier feature encoder (Tancik et al., 2020), followed by an MLP. The performance of these architectures using the DiT model are given in Table 6a. Note that for the DiT model, the offset architecture works the best. However, as shown in Table 6b, the cross-attention adapter works better for text-to-image models such as Stable Diffusion (Rombach et al., 2022). Hence, we used the offset architecture for class-conditional generation, and the cross-attention adapter for more complex text-to-image models. We also experimented with using dropout in the offset MLPs for further regularization but found that the model performs best without using any dropout (see Table 7).

**Adapter initialization**   While adapters are typically initialized with zero values such that $\boldsymbol{\epsilon}_{[\boldsymbol{\theta}, \boldsymbol{\psi}]} = \boldsymbol{\epsilon}_{\boldsymbol{\theta}}$ at initialization (Houlsby et al., 2019), Table 8 shows that Xavier initialization yields better results for guidance distillation. Therefore, we recommend avoiding zero initialization of the adapters for AGD.

**Dimensionality of the adapter**   We now examine the impact of adapter dimensionality in Table 9. Our results show that increasing the hidden dimension initially improves FID but eventually leads to degradation, likely due to overfitting. Therefore, we recommend designing adapters with fewer than 5% additional parameters w.r.t. the base model.

**Training loss functions**   We also explored various loss functions for training AGD. Specifically, we experimented with $\ell_1(\mathbf{x}, \mathbf{y}) = \|\mathbf{x} - \mathbf{y}\|_1$ and a weighted $\ell_2(\mathbf{x}, \mathbf{y}) = \lambda(t)\|\mathbf{x} - \mathbf{y}\|_2^2$, where $\lambda(t)$ is a weighting function depending on the time step. As shown in Table 10, the simple $\ell_2$ loss with $\lambda(t) = 1$ performs best.

## C   Details of the evaluation samples for qualitative comparison

**DiT**   The samples used a guidance scale of 4.

Table 6: Ablation study on adapter architectures across different baseline models. We observe that offset adapters perform best for class-conditional models, while cross-attention adapters are more effective for text-to-image generation.

(a) DiT.

| Architecture | FID ↓ | Prec. ↑ | Recall ↑ |
|---|---|---|---|
| Cross-attention | 5.49 | **0.83** | 0.65 |
| Offset | **5.03** | 0.80 | **0.68** |
| Positional encoding | 5.25 | 0.81 | 0.66 |
| Gating | 5.54 | **0.83** | 0.66 |

(b) SD2.1

| Architecture | FID ↓ | Prec. ↑ | Recall ↑ |
|---|---|---|---|
| Cross-attention | **21.09** | **0.66** | **0.55** |
| Offset | 22.05 | 0.63 | 0.54 |

Table 7: Ablation on the dropout rate using DiT.

| Dropout | FID ↓ | Prec. ↑ | Recall ↑ |
|---|---|---|---|
| 0% | **5.22** | 0.80 | **0.67** |
| 10% | 5.27 | 0.82 | **0.67** |
| 20% | 5.39 | 0.83 | 0.66 |
| 50% | 5.69 | **0.85** | 0.63 |

Table 8: Ablation on the initialization type of the adapter layers.

| Init. scheme | FID ↓ | Prec. ↑ | Recall ↑ |
|---|---|---|---|
| Zero | 5.24 | **0.81** | **0.68** |
| Xavier | **5.03** | 0.80 | **0.68** |

Table 9: Ablation on the hidden dimensionality of the adapters.

| Dim. | Params | FID ↓ | Prec. ↑ | Recall ↑ |
|---|---|---|---|---|
| 64 | 0.8% | 5.33 | **0.81** | 0.67 |
| 128 | 2.5% | **5.03** | 0.80 | **0.68** |
| 256 | 6.1% | 5.22 | 0.80 | 0.67 |
| 512 | 17.2% | 5.26 | **0.81** | 0.67 |

Table 10: Effect of using different loss functions for distillation.

| Loss | Weight $\lambda(t)$ | FID ↓ |
|---|---|---|
| $\ell_1$ | 1 | 9.91 |
| $\ell_2$ | 1 | **5.03** |
| $\ell_2$ | $\sigma(t)$ | 5.30 |
| $\ell_2$ | $\frac{1}{2}\|1 - \cos\angle(\tilde{\epsilon}_{\boldsymbol{\theta}}, \epsilon_{\boldsymbol{\theta}})\|$ | 6.64 |

**SD2.1** The samples used a guidance scale of 10. From left to right, the prompts used in Figure 5b were:

1. "A cat on the flower."

2. "A close-up of a blooming flower."

3. "A quiet beach at sunset with gentle waves."

4. "A calm lake reflecting the blue sky."

**SDXL** The samples used a guidance scale of 12. Further, the prompts used in Figure 5c were:

1. "A modern reinterpretation of a classical Renaissance painting, where futuristic elements and digital motifs merge with traditional portraiture."

2. "A fantastical scene of a celestial garden floating in space, featuring luminous, otherworldly flora against a backdrop of swirling galaxies."

3. "A hyper-realistic digital painting of a futuristic metropolis at sunset, with neon lights reflecting off rain-soaked streets and towering holograms."

4. "A cozy winter scene of a remote mountain village, with softly glowing windows, snow-covered rooftops, and a star-filled night sky."

# D    Additional implementation details

Section 4.2 provides the main implementation details. The DiT-XL/2 model was trained with a batch size of 64 for 5000 gradient steps, the SD2.1 model with a batch size of 8 for 5000 gradient steps, and the SDXL model with a batch size of 1 for 20000 gradient steps. These settings were selected based on the maximum batch size that fits within 24,GB of VRAM. For all quantitative experiments, we set the guidance scale to the value that achieved the best FID for each method. The AGD implementation will be publicly released to support further research on guidance distillation.

The FID scores for class-conditional models were computed using 10k generated samples and the entire ImageNet training set. For text-to-image models, we used the full COCO-2017 validation set as the real data. All metrics were computed using the ADM evaluation code base (Dhariwal & Nichol, 2021) to ensure fairness across experiments.

# E    Additional visual samples

Figures 10 to 21 provide additional visual samples comparing AGD and CFG across various models used in this work. Similar to our main findings, AGD samples consistently match or surpass CFG samples in both quality and diversity. The results are best seen when zoomed in.

CFG    AGD (Ours)

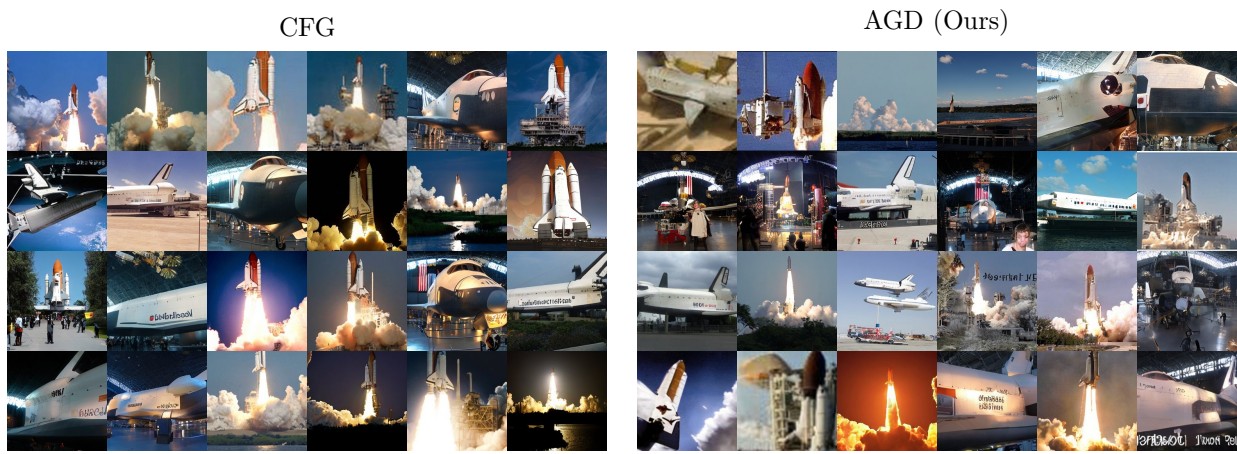

Figure 10: Uncurated samples using DiT-XL/2. Class label: "Space shuttle" (812), guidance scale: 2.

CFG    AGD (Ours)

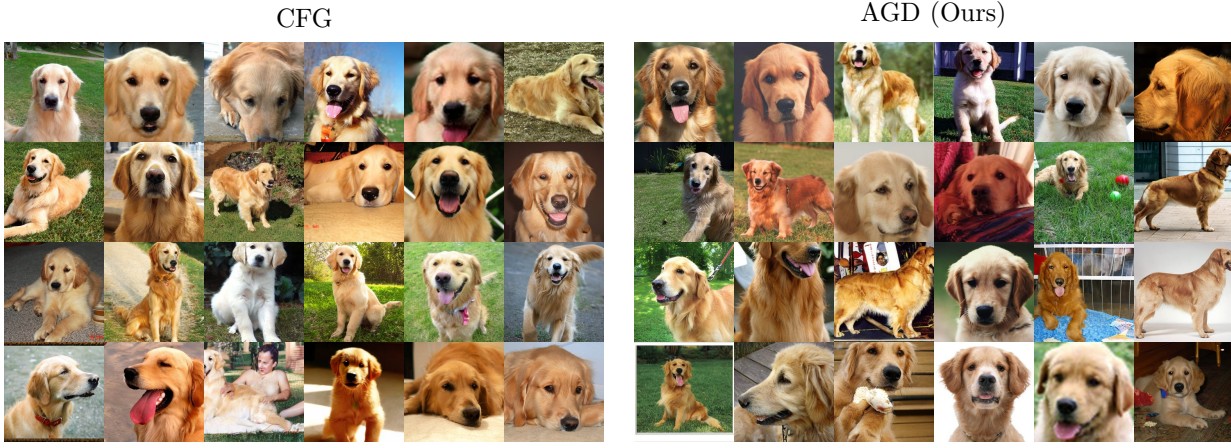

Figure 11: Uncurated samples using DiT-XL/2. Class label: "Golden retriever" (207), guidance scale: 3.

CFG                                                    AGD (Ours)

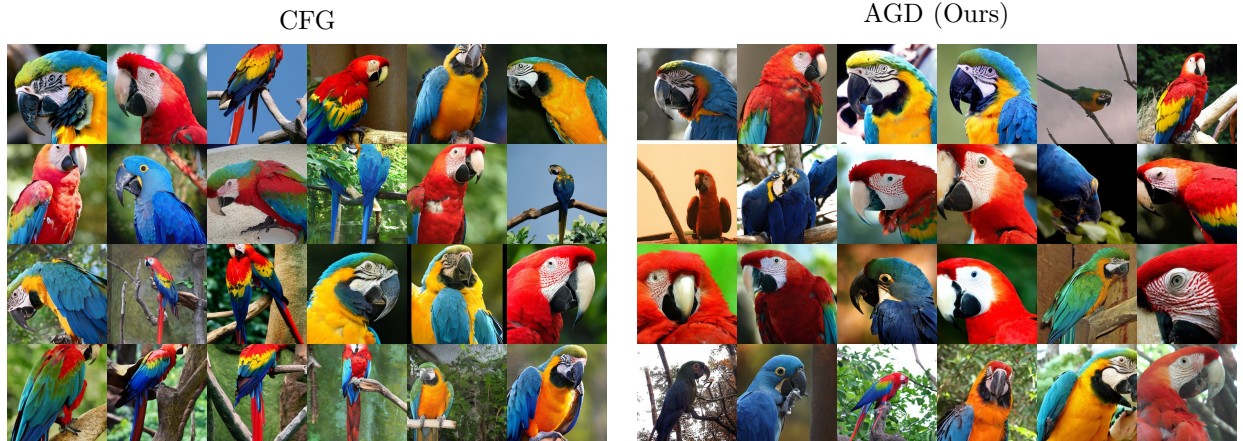

Figure 12: Uncurated samples using DiT-XL/2. Class lable: "Macaw" (88), guidance scale: 3.

CFG                                                    AGD (Ours)

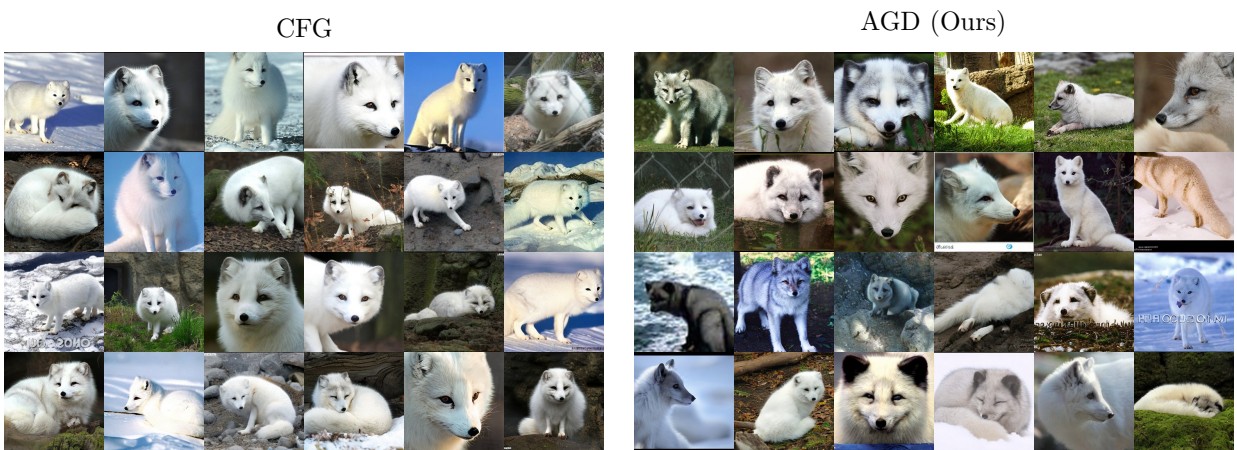

Figure 13: Uncurated samples using DiT-XL/2. Class label: "Arctic fox" (279), guidance scale: 4.

CFG                                                    AGD (Ours)

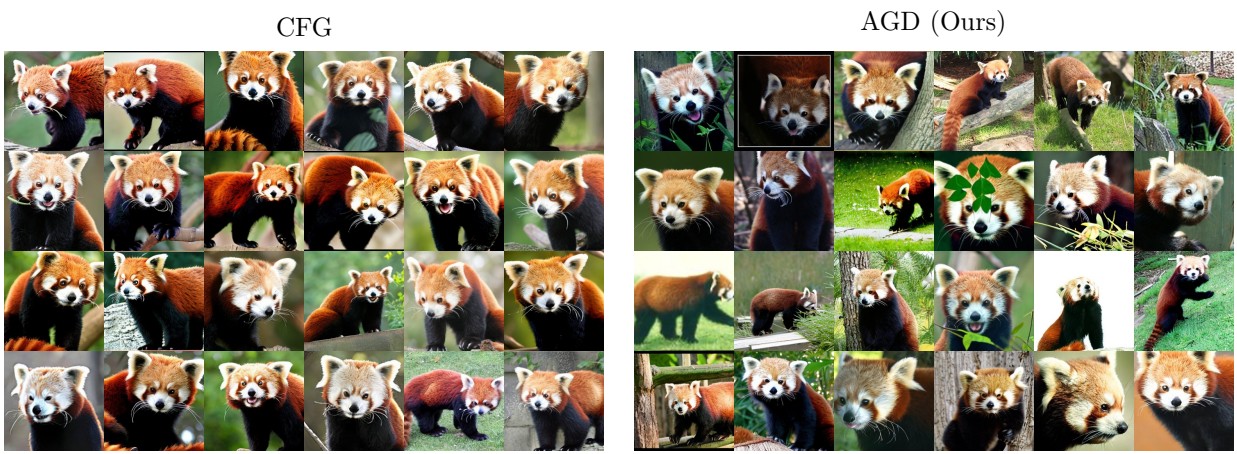

Figure 14: Uncurated samples using DiT-XL/2. Class lable: "Red panda" (387), guidance scale: 5.

CFG                                           AGD (Ours)

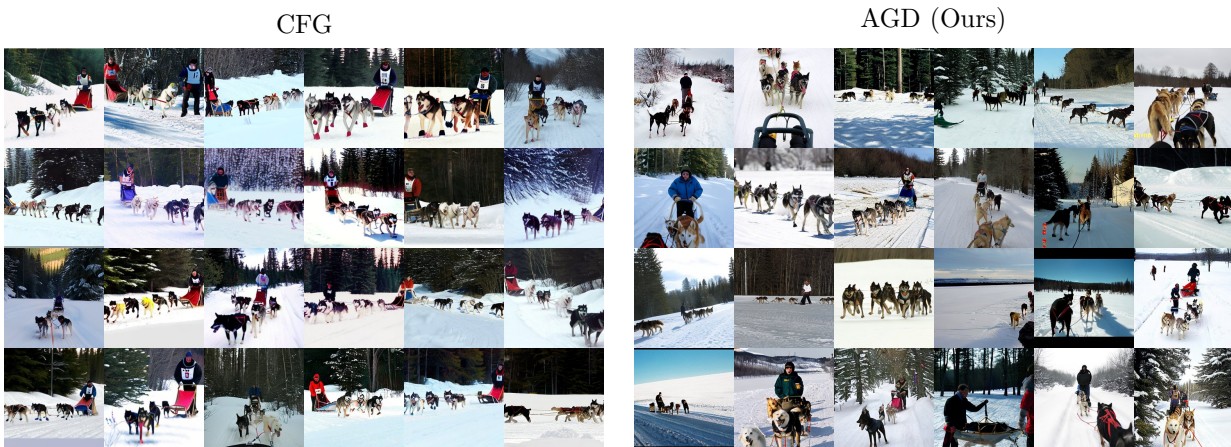

Figure 15: Uncurated samples using DiT-XL/2. Class label: "Dog sled" (537), guidance scale: 6.

CFG                                           AGD (Ours)

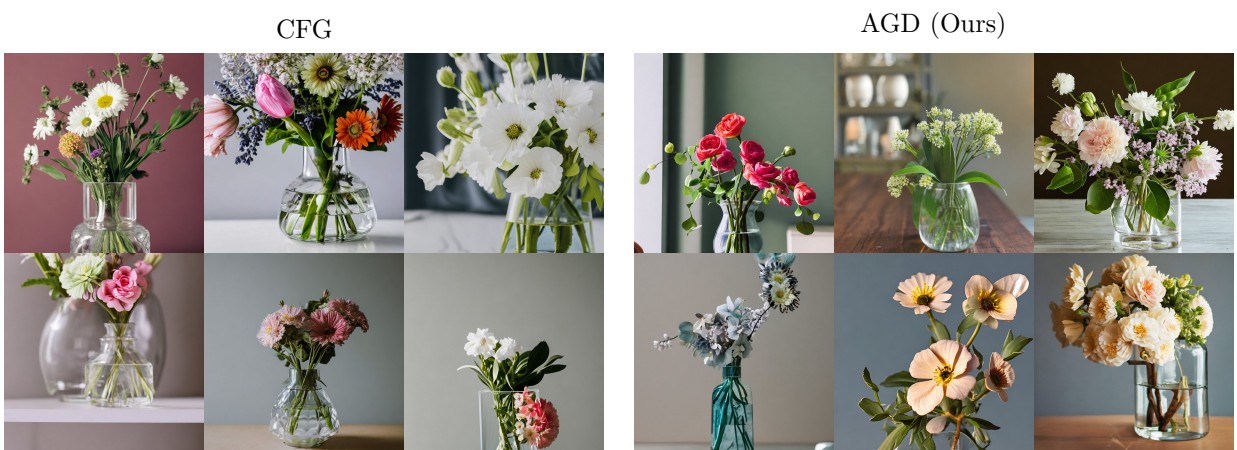

Figure 16: Uncurated samples using SD2.1. Prompt: "A close up of a clear vase with flowers.", guidance scale: 10.

CFG                                           AGD (Ours)

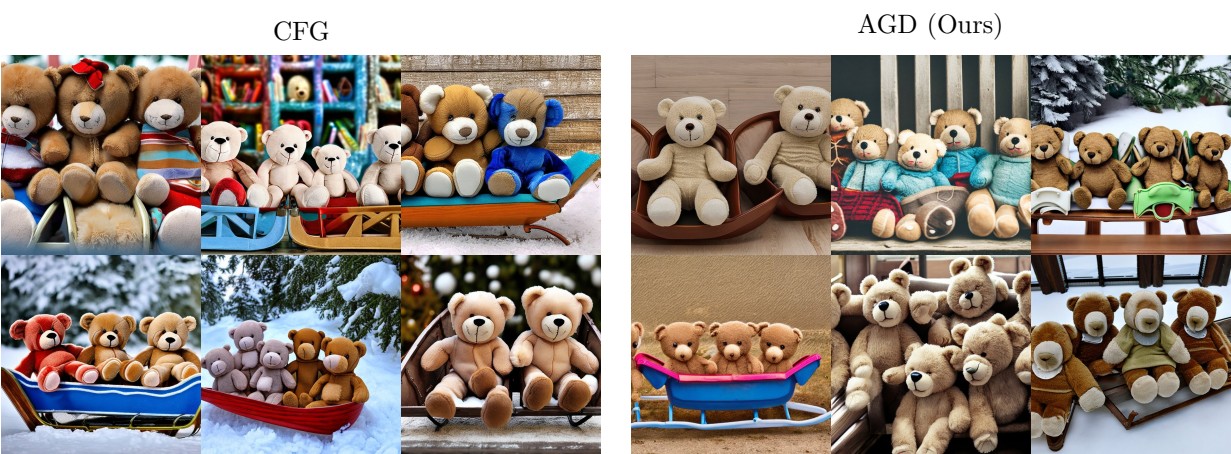

Figure 17: Uncurated samples using SD2.1. Prompt: "A set of plush toy teddy bears sitting in a sled.", guidance scale: 10.

CFG

AGD (Ours)

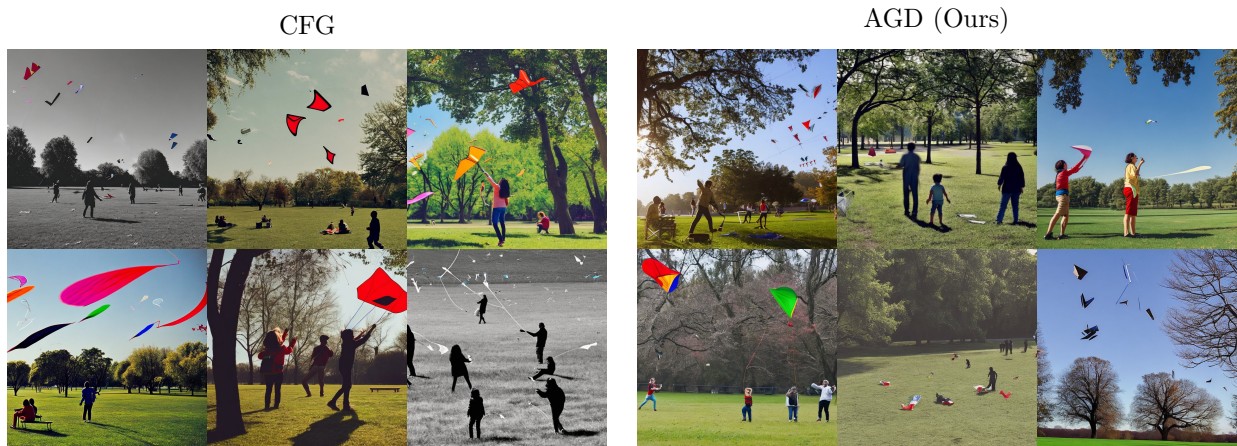

Figure 18: Uncurated samples using SD2.1. Prompt: "People flying kites in a park on a windy day.", guidance scale: 10.

CFG

AGD (Ours)

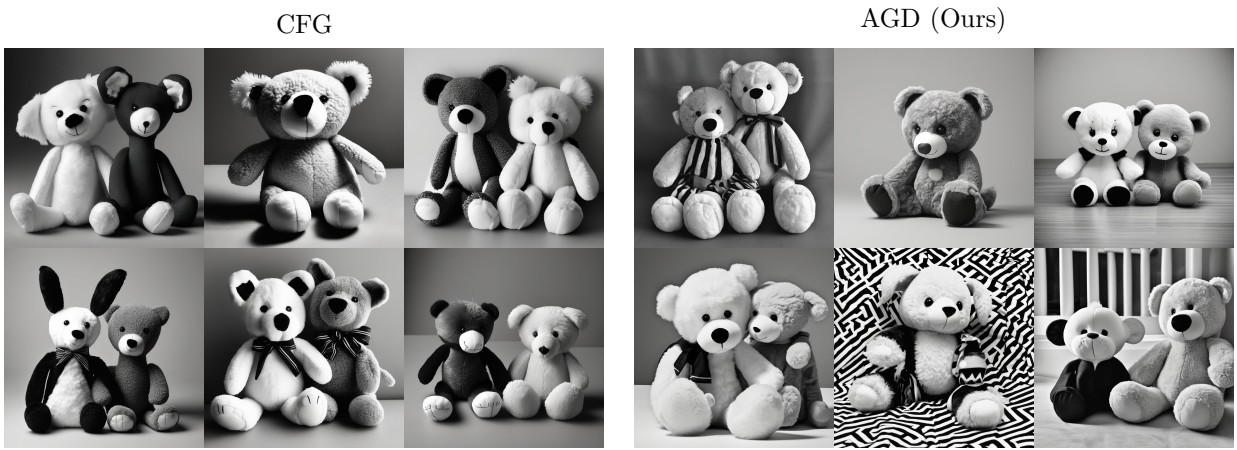

Figure 19: Uncurated samples using SDXL. Prompt: "Two stuffed animals posed together in black and white." guidance scale: 12.

CFG

AGD (Ours)

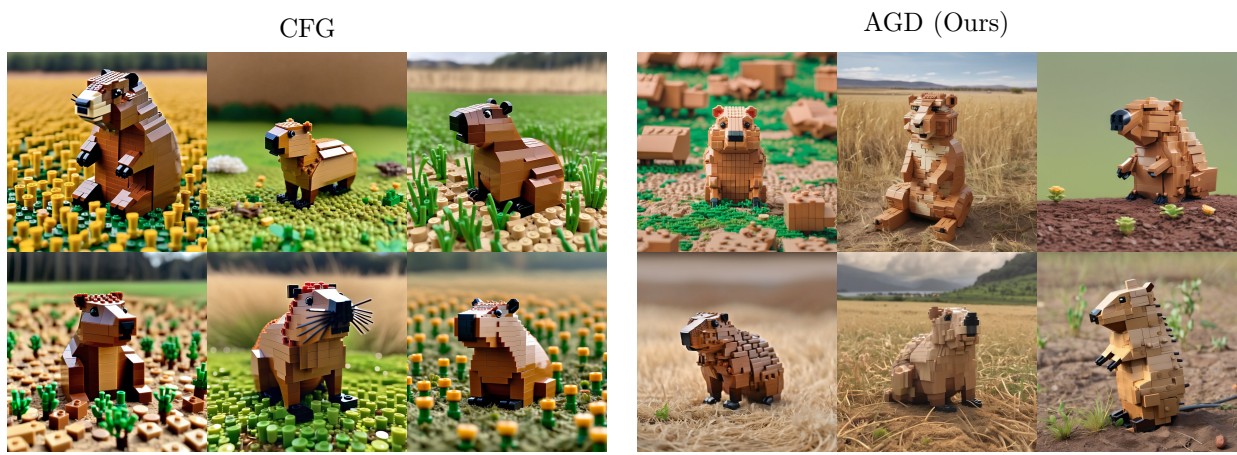

Figure 20: Uncurated samples using SDXL Prompt: "A capybara made of lego sitting in a realistic, natural field.", guidance scale: 12.

CFG AGD (Ours)

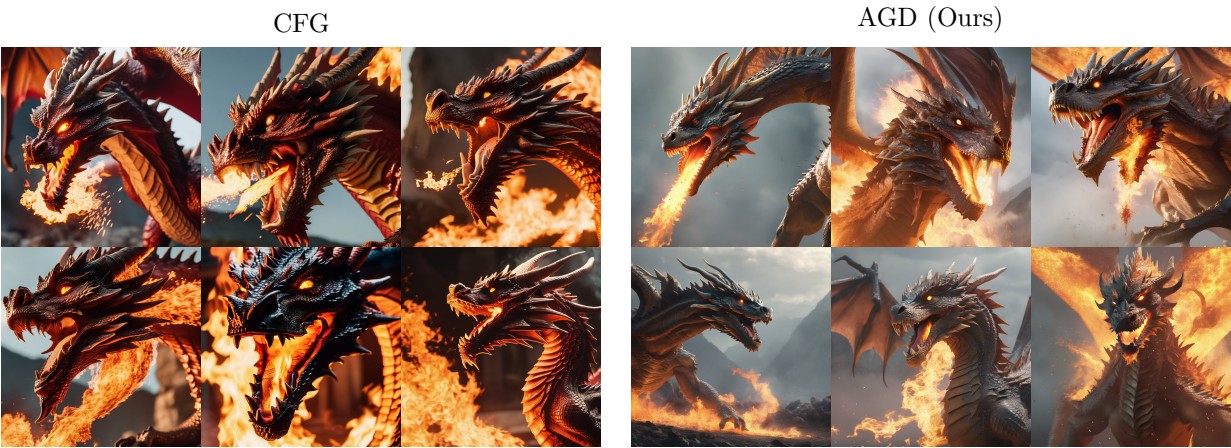

Figure 21: Uncurated samples using SDXL Prompt: "A close-up of a fire spitting dragon, cinematic shot.", guidance scale: 12.

