# OpenReview forum: "Efficient Distillation of Classifier-Free Guidance using Adapters"
_TMLR — Accepted by TMLR_

### Review · Reviewer_j5zt · 2025-08-27

**Summary Of Contributions:**

This paper focuses on a potential issue in the classifier-free guidance (CFG) ; that is, it requires two forward passes during the inference step. To solve this issue, this paper proposes the adapter guidance distillation (AGD), an adapter-based approach to mimic the behavior of the CFG procedure. Also, this paper proposes an approach to train this AGD and empirically verifies the performance of AGD.

Strength:

I believe the proposed approach has (at least partially) addressed the issue mentioned in the beginning of this paper. The AGD doesn't require to run two forward passes to follow the guidance.

Weakness:

However, the proposed method has several fundamental issues: (1) This paper doesn't consider the additional time cost for training the adapter. (2) I would also be concerned about the generalization ability of the adapter for those prompts that are not distilled.

**Audience:**

Yes

**Audience Explanation:**

I don't work on the diffusion model but I am sure that the method proposed in this paper would be interesting for those who are working in this field. Because in traditional methods (e.g. CFG), it requires two forward passes, this paper successfully reduces the number from 2 to 1. This result might be surprising for people in this field as it will violate the common intuition.

**Broader Impact Concerns:**

I don't have any ethical concern on this work.

**Claims And Evidence:**

No

**Claims Explanation:**

Here I further explain the two weaknesses I mentioned above, as the current results can only partially support the claims made in this paper.

1. As I commented above, trainning the adapter requires additional time or training resources. These cost should also be considered when cmpared with the CFG method. As the claim made in this paper is that the AGD will save the computational overhead in the second paragraph (page 1), (from my perspective), it should be the entire computational overhead. So, the current result is not enough to support this claim.

2. The original diffusion model typically was trained on a huge dataset containing millions of prompts, whereas distillation training typically relies on only hundreds or at most a few thousand captions. Then the evaluation is made over similar captions or prompts. This result is not sufficient to support the effectiveness of the proposed AGD method as it is hard to tell how the adapter will perform when it recieves a prompt that has not been used during the distillation training procedure. The adapter will perform significantly different from CFG as it has not been "trained" to be the same.

As a result, I would suggest add some additional experiments to support these claims (see Requested Changes).

**Requested Changes:**

1. I suggest add the inference time cost of both CFD and AGD method (for example, CFD takes 2 seconds and AGD takes 1 second). Also, add the time cost of training the adapter (e.g. 100 seconds). Then the result tells that after generating 100 batches of images, using AGD will alway lead to positive gains.

2. I suggest evaluate the generalization ability in a larger dataset. For example, evaluate the DiT for the whole ImageNet using CFG. But train AGD on a few class of it. When the training data contains the whole ImageNet, I would expect to see the performance would be similar to the CFG. But when it only uses a few classes, the performance might be slightly worse.

Also, the experiment section is not very clear. There are multiple sizes of DiT models. Which one are you using?

---

> ### Author Response · Authors · 2025-08-28
>
> We thank the reviewer for their time and for providing feedback on our submission. Below, we address the concerns in more detail and are happy to discuss further.
>
> ### **Training time for distillation**
> Our claim specifically concerns *inference-time* overhead. Training the distillation component is fast and needs to be done only once. Given the widespread use of diffusion models, the inference speed-up easily justifies this one-time training cost. As reported in the paper, the DiT model can be distilled with AGD in 33.9 minutes (equivalent to generating ~170 images, given DiT’s 12-second per-image inference time). Once distilled, it provides *twice the sampling speed* of guided models for all subsequent generations. In other words, the nominal cost of distillation more than pays for itself through downstream savings during inference. This approach is also standard practice in recent models such as Flux [1], where a pretrained model is tuned via guidance distillation specifically to accelerate inference. Our claim that inference is 2$\times$ faster than CFG is supported by the numbers presented in Section 5.9.
>
> ### **Generalizability of adapters**
> All samples shown in the paper use prompts not seen during training. For quantitative evaluation, the training and validation captions for the Stable Diffusion models are disjoint. We believe this demonstrates the generalizability of our method. Furthermore, prior step-distillation work (e.g., adversarial distillation [2]) shows that fine-tuning on a relatively small set of captions yields good generalization for distillation purposes. This is because the base model is kept frozen and only a small set of parameters on top is trained, which leads to fast training and strong generalization. We would be happy to include additional visual results and experiments to further support this point.
>
> ### **Other questions**
> The inference and training costs for DiT are mentioned in Section 5.9. We use DiT-XL/2 at 256$\times$256 resolution. We will clarify this in the revised version of the paper, and we can also provide inference times for Stable Diffusion models.
>
> [1] [https://bfl.ai/](https://bfl.ai/)
>
> [2] Sauer A, Boesel F, Dockhorn T, Blattmann A, Esser P, Rombach R. Fast high-resolution image synthesis with latent adversarial diffusion distillation. In *SIGGRAPH Asia 2024 Conference Papers*. 2024 Dec 3 (pp. 1–11).

---

> > ### Comment · Reviewer_j5zt · 2025-09-03
> >
> > Thanks for the clarification! I didn't locate these information before and I would believe it has been sufficiently clear to me. To this reason, I willto accept.

---

### Review · Reviewer_87Q3 · 2025-08-31

**Summary Of Contributions:**

In this paper the authors introduce the method of adapter guidance distillation (AGD) as a substitute of classifier-free guidance (CFG) methods for conditional diffusion models. While CFG remains a critical method for increasing the output quality of diffusion models, it is known to be very costly due to the inherent need to double the sampling cost as guidance. AGD is proposed as an alternative technique that simulates CFG while greatly reducing the additional cost of sampling, while maintaining the improved level of output quality.

The paper's main contribution is the aforementioned introduction of AGD, which utilizes 'adapters', lightweight modules that augments a frozen base model, to greatly reduce the cost of inference and simulate CFG with only a single pass without needing to fine-tune the entire, possibly complex model. Additionally, they identify a previously unrecognized observation that CFG and similar methods introduce a significantly differing training trajectory than classical diffusion, and thus proposes training on CFG-guided trajectories as opposed to classical ones. They claim that AGD can be deployed and trained on small-scale processing units such as common household ones and do not require much processing power.

The authors perform comprehensive experimental evaluations of AGD against traditional CFG as well as guidance distillation (GD) without adapters. They claim that the experimental results showcase that AGD matches and possibly improves on the quality of the generated output, while greatly improving the efficiency and cost of training.

**Audience:**

Yes

**Audience Explanation:**

Diffusion models and their application in text/image generation is a vital and vigorous topic in machine learning and large language models. Accompanied with the interests are a variety of concerns and critiques of their ethical concerns and environmental impacts. This work addresses these interests and concerns, by proposing a novel technique that can help diffusion models deploy on smaller scale hardware that consumes significantly less energy and resources, both material and time-wise. As a result, I think the contributions of this paper would be of interest to TMLR's audience.

**Broader Impact Concerns:**

Ethical concerns on large language models, generative artificial intelligence, and diffusion models mostly focus on (1) the energy consumption and environmental impact of training and deploying these models, and (2) the fair sourcing and usage of training data and output. The authors' work intrinsically addresses the first point. While I do not believe it precisely the authors' responsibility to address the second, I would appreciate some additional sources and citations that discuss this aspect in more depth; The reference provided is published in 2021, while the landscape of AI and diffusion models have altered significantly since then.

**Claims And Evidence:**

Yes

**Claims Explanation:**

The claims made in the submission are sound and supported by evidence to my best knowledge.

The authors claim that AGD produces outputs that are "more visually appealing" and "often have better visual coherence", which is not exactly simple to quantify. The other claims, including that AGD produces images of comparable quality and that AGD trains more efficiently, is supported by experimental evidence.

I am not familiar with diffusion models and their theoretical foundations, and as such cannot claim for certain that the evidences and analyses provided are completely reasonable; however I am able to follow the intuitions behind the authors' ideas and tend to believe that their claims are sound.

**Requested Changes:**

None. Unfortunately I do not believe myself informed enough about the topic beyond a general audience level to propose reasonable inputs.

---

> ### Author Response · Authors · 2025-09-03
>
> We thank the reviewer for the comments and for supporting our work. We would like to clarify that our claims about being more visually appealing refer specifically to the figures presented in the qualitative results of the paper. Regarding the ethics question, we would like to update the reference to [1], which covers the current landscape of generative models. We would also be happy to engage in further discussion if needed.
>
> [1] Lin X, Losavio M. A Comprehensive Survey on Bias and Fairness in Generative AI: Legal, Ethical, and Technical Responses.

---

### Review · Reviewer_QNzb · 2025-09-02

**Summary Of Contributions:**

**Summary**

This paper proposes a method based on distillation to substitute classifier-free guidance (CFG) in diffusion models. In particular, CFG requires at each diffusion step two calls to a denoising model, one to the conditioned model and one to the unconditioned one. Here, the model is trained to reproduce CFG denoising trajectories by simply adding a small model, called adapter, which purpose is to steer the base denoising model. The main differences with previous work (Meng et al.) are that (i) the small adapters add negligible memory and runtime overhead, and (ii) the adapters are trained on CFG-guided trajectories instead of unconditional diffusion trajectories, which offers a boost in the generation quality. The method is validated three base diffusion models, using FID as the main quantitative metric.


**Strengths:**

- The paper is well-written and easy to follow, even for people outside of the subfield. The method, albeit simple, is explained very clearly.
- The presented results seem to be promising, as the proposed AGD method offers quality comparable to CFG while halving the calls to the denoising models.
- Moreover, AGD offers better generation quality and lower memory requirements compared to guidance distillation (Meng et al.).

**Weaknesses:**

- For the larger diffusion models, ADG seems to produce slightly worse FID scores compared to CFG, which might limit the applicability of the method when high-quality images are required. This, however, is quite expected, and not a major issue.
- In Table 1, it would be interesting to report inference times, as reporting solely the calls to the denoiser is not that informative. In Section 5.9 the times for CFG are not reported, which is a bit suspicious.
- The code is not available to reviewers.

Unfortunately, I remark that this paper is significantly outside my area of expertise, so I could not check the details of the paper thoroughly.

**Audience:**

Yes

**Audience Explanation:**

I believe that a large portion of the diffusion models subfield will find this paper very interesting, including but not limited to TMLR's audience.

**Broader Impact Concerns:**

Section A in the Appendix is sufficient.

**Claims And Evidence:**

Yes

**Claims Explanation:**

The method is validated on multiple models and several ablation studies are presented. Therefore, I find the experimental evidence quite convincing.

**Requested Changes:**

- Please provide inference times in Table 1. I find this quite critical for assessing the paper's contribution.
- Also, please provide the code to reviewers (and make it public after acceptance!).

---

> ### Author Response · Authors · 2025-09-03
>
> We thank the reviewer for the detailed comments and for supporting our work. Please find our responses to each comment below, and we would be happy to engage in further discussion.
>
> ### **Code request**
> We agree with the reviewer that sharing the code is an important aspect of our work that will help the community in developing further methods for guidance distillation. Please find the anonymous version of our code at [the following URL](https://anonymous.4open.science/r/agd). We will publish the full repository after the review process.
>
> ### **Inference time for Table 1**
> Since adapters introduce negligible overhead compared to the base model (as shown in the inference time for DiT in Section 5.9), the NFEs directly translate to inference time for our method (i.e., 2× NFEs means 2× inference time). This is also why we did not report CFG inference time in Section 5.9 (as it is 2× the unguided sampling, or equivalently ~12 sec/sample). That said, we thank the reviewer for raising this question, and we will provide inference times for all models in terms of throughput to further support our claim.
>
> ### **Comment about FID**
> Please note that FID is most accurate for class-conditional ImageNet generation, where we show that AGD even slightly outperforms CFG. For text-to-image models, while large differences in FID indicate clear quality gaps, FID is not very sensitive to small-scale differences (e.g., 22.82 vs. 22.98). As such, visual inspection is often the most natural way to compare the quality of such setups. As shown in the qualitative results, AGD performs very similarly to CFG, which further supports our claim.

---

> > ### Comment · Reviewer_QNzb · 2025-09-04
> >
> > Thanks for your reply. Given that you provided the code and the clarification about inference times, I have no further concerns.

---

### Decision · Action_Editor_dzkR · 2025-09-30

**Recommendation:** Accept with minor revision

**Additional Comments:**

In the revised version, I encourage the authors to include inference times for all models in terms of throughput, and to clearly state that the DiT experiments are conducted with DiT-XL/2 at 256×256 resolution. It would also be helpful to provide corresponding inference times for the Stable Diffusion models.

**Audience:**

Yes

**Audience Explanation:**

The topic of diffusion models is highly relevant to TMLR’s audience. The paper's contributions are likely to attract interest from researchers working on generative models and their applications.

**Claims And Evidence:**

Yes

**Claims Explanation:**

The submission provides solid experimental evidence, including validation on multiple models and ablation studies, which overall support the authors’ claims. While some aspects (e.g., “visual appeal”) are harder to quantify, the core claims on quality and efficiency are backed by convincing results and presented clearly.